# SELF-GUIDE: Better Task-Specific Instruction Following via Self-Synthetic Finetuning

**Chenyang Zhao**[1]\*, **Xueying Jia**[2]\*,
**Vijay Viswanathan**[2], **Graham Neubig**[2], **Tongshuang Wu**[2]
[1]Department of Computer Science and Technology, Tsinghua University, Beijing, China
[2]Language Technologies Institute, Carnegie Mellon University, Pittsburgh, PA, USA
zhaochenyang20@gmail.com
{xjia2,vijayv,gneubig,sherryw}@andrew.cmu.edu

## Abstract

Large language models (LLMs) hold the promise of solving diverse tasks when provided with appropriate natural language prompts. However, prompting often leads models to make predictions with lower accuracy compared to finetuning a model with ample training data. On the other hand, while finetuning LLMs on task-specific data generally improves their performance, abundant annotated datasets are not available for all tasks. Previous work has explored generating task-specific data from state-of-the-art LLMs and using this data to finetune smaller models, but this approach requires access to a language model other than the one being trained, which introduces cost, scalability challenges, and legal hurdles associated with continuously relying on more powerful LLMs. In response to these, we propose SELF-GUIDE, a multi-stage mechanism in which we synthesize task-specific input-output pairs from the student LLM, then use these input-output pairs to finetune the student LLM itself. In our empirical evaluation of the Natural Instructions V2 benchmark, we find that SELF-GUIDE improves the performance of LLM by a substantial margin. Specifically, we report an absolute improvement of approximately 15% for classification tasks and 18% for generation tasks in the benchmark's metrics. This sheds light on the promise of self-synthesized data guiding LLMs towards becoming task-specific experts without any external learning signals.[1]

## 1 Introduction

One of the traits of large language models (LLMs) that has captured the imagination of model developers is their potential to automate a broad range of highly complex tasks with relative ease (Brown et al., 2020a). Where previous generations of models required vast task-specific training sets, LLMs now offer the promise of enabling similar accuracy by simply writing a prompt and supplying a few examples. However, this may be a *false promise*. Prompting typically leads models to make predictions with lower accuracy compared to finetuning a model with ample training data (Gao et al., 2021; Zhang et al., 2023). Moreover, LLMs' performance crucially depends on their ability to *follow instructions* outlined in the prompts and even minor alterations to these prompts can result in a notable performance decline (Sclar et al., 2023a).

On the other hand, in data-abundant tasks, finetuning a pre-trained language model through supervised finetuning (Ouyang et al., 2022; Wei et al., 2022) and reinforcement learning from human feedback (Lambert et al., 2022) has proven a successful strategy. However, the effectiveness of this approach diminishes notably for underrepresented tasks suffering from

---

\*Equal contribution

[1]All code and data necessary to reproduce experiments are released publicly: https://github.com/zhaochenyang20/Prompt2Model-Self-Guide

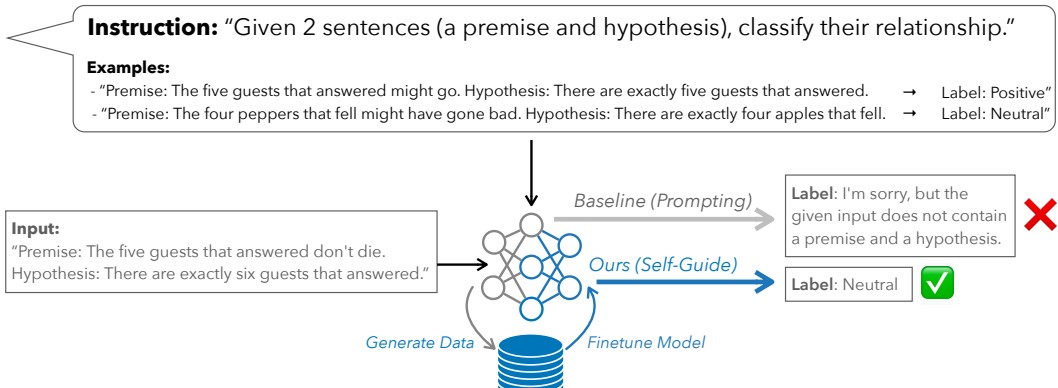

Figure 1: SELF-GUIDE uses a model's ability to generate synthetic data as a vehicle to improve the model's ability to execute a task as specified by an instruction.

data scarcity (Li et al., 2022), highlighting the critical need for high-quality training data. To this end, recent studies have explored the potential of utilizing potent "teacher" LLMs to create task-specific training data, thereby enhancing the performance of comparatively less advanced "student" LLMs (Khattab et al., 2023; Viswanathan et al., 2023). While this strategy is effective, its feasibility is hampered by the cost, scalability, and legal hurdles associated with the continuous reliance on more powerful teacher LLMs. And when there isn't a stronger model available to provide learning signals, this approach is fundamentally infeasible.

In this paper, we instead propose to finetune language models on self-synthesized data to improve their ability to follow instructions on *specific tasks* with *minimal annotated data*. Concretely, we ask the question: can we enhance a model's performance for arbitrary tasks without external training signals, such as labeled data or another teacher model? In our affirmative answer to this question, we introduce SELF-GUIDE, a novel methodology that enables LLMs to better execute task-specific instructions without requiring additional data or training signals. SELF-GUIDE operates in the *few-shot setting*, where we are given only a task instruction and up to three examples of task demonstrations. SELF-GUIDE works by first employing the target model to generate a synthetic dataset for a given task. The model is then finetuned on this "self-generated" data.

This approach differs from previous methods for bootstrapping a model from its own outputs. Self-Instruct (Wang et al., 2023) is the most similar work; here, they use a base LLM (GPT-3) to generate a synthetic instruction-following dataset to finetune the same LLM. In contrast to their method, which aims to improve general-purpose LLM capabilities, our method aims to optimize an LLM for a specific task instruction. Methodologically, Self-Instruct generates a large set of instructions and demonstrations to use for instruction-finetuning. While Self-Instruct asks an LLM to self-generate synthetic demonstrations for each synthetic instruction, their method only generates a single demonstration for each instruction, while our method can effectively self-generate hundreds of examples for a given instruction. Our methods are complementary. We find that our method can perform target tasks with significantly greater reliability when applied on top of general-purpose instruction finetuned models (e.g. the kind of model resulting from Self-Instruct). We provide more details regarding other related methods in Section 5.

In our empirical evaluation of SELF-GUIDE using on multiple tasks from Super-NaturalInstructions V2 (Wang et al., 2022), applying SELF-GUIDE to an already instruction-tuned model (Vicuna-1.5-7B, (Zheng et al., 2023)) yields an absolute performance improvement of **17.9 points** of ROUGE-L for open-ended generation tasks and **14.5 points** of accuracy on classification tasks, compared against the baseline of prompting the same model with the same prompt.

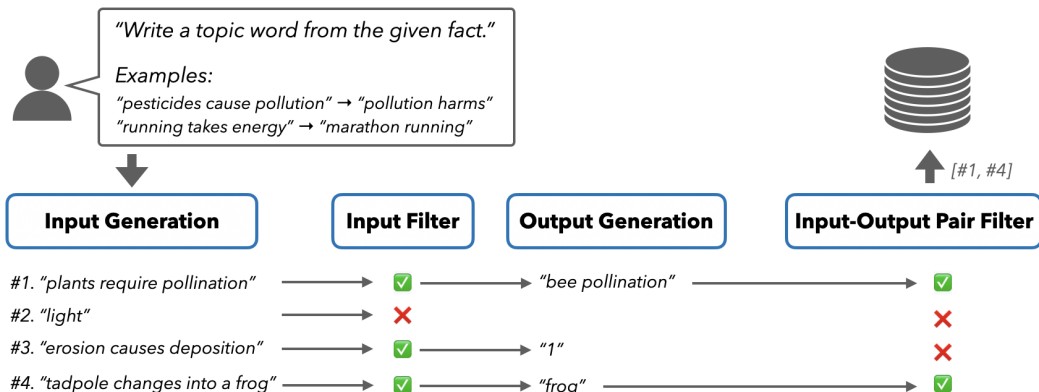

Figure 2: At the heart of SELF-GUIDE lies an efficient and effective multi-stage generation mechanism, where the LM generates input-output pairs step by step. After the generation and filtering, the self-generated data are further used to finetune the LM itself. This figure describes the process for the generation tasks.

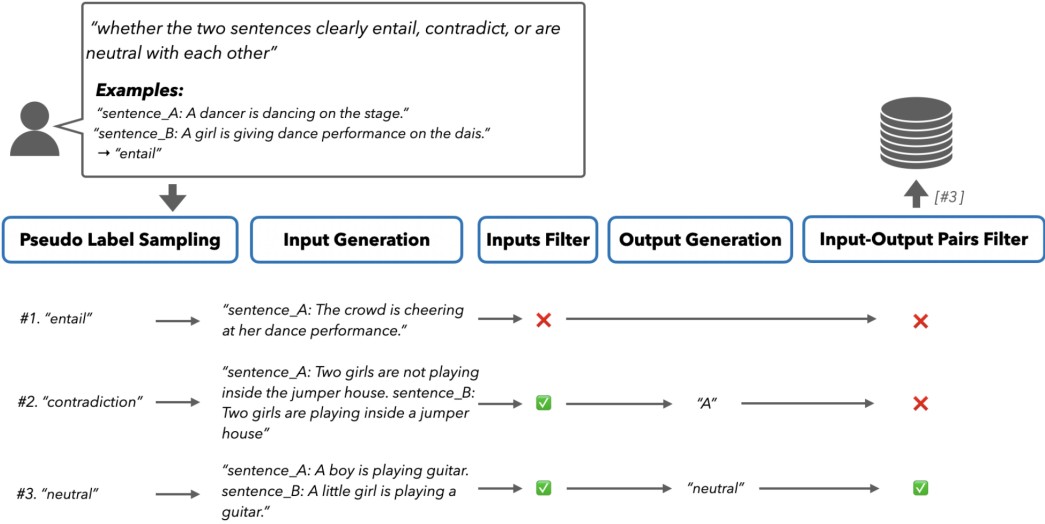

Figure 3: This figure describes the process for the classification tasks; we use a slightly modified procedure for self-generating data for classification tasks. Put simply, we first generate pseudo-labels, then generate corresponding diverse inputs, and finally generate true labels. Regarding the Input-Output Pairs Filter, a set of labels will be provided to filter out labels. Further details will be described in Section 2.1.

## 2 SELF-GUIDE

In the following section, we outline the proposed SELF-GUIDE framework (illustrated in Figure 2 and Figure 3). As input, SELF-GUIDE takes an instruction (e.g. "Write a topic word from a given fact."), and a few example inputs and outputs (e.g. "pesticides cause pollution" as input fact, "pollution harms" as output topic word). Our method proceeds in multiple stages, where each stage performs procedures to improve quality, either through rule-based filters on data quality, or hyperparameter search. Below, we describe the key design points.

## 2.1 Data Generation

**Input Generation**    The input generation process starts by extracting inputs from provided example pairs and combining them with the instruction to populate a prompt template. The constructed prompts are then forwarded to the LLM to obtain model-generated inputs. After each round of prompting, the newly generated inputs are added to an input repository. A subset of inputs is randomly sampled from this repository and merged with inputs from the initial examples to formulate a new prompt, aiming to incrementally expand the set of LLM-generated inputs. We only go through one round of input generation and subsequently, in the quality refinement stage, rule-based filters are applied to remove low-quality inputs. Detailed descriptions of this quality refinement process are provided in Section 2.2.

The prompt templates employed vary based on the task type-generation or classification. For generation tasks, a simple prompt template instructs the model to generate a new input following the provided examples. In contrast, in classification tasks, the model is directed to generate an input corresponding to a randomly chosen label from the available set, allowing for a more balanced label distribution in the generated examples. Further details on the prompt templates are provided in Appendix A.1.1.

**Output Generation**    The output generation employs conventional in-context learning techniques (Min et al., 2022a; Brown et al., 2020b). It provides the model with instructions and original examples, allowing it to annotate every single input that was generated earlier in the input generation stage. After obtaining all annotated outputs for the previously generated inputs, another round of rule-based filtering takes place to select the final synthetic dataset, the details of which are described in Section 2.2. The comprehensive prompt templates used for output generation are provided in Appendix A.1.2.

## 2.2 Quality Optimization: Temperature and Rule-based Filters

The quality of generated data is essential for the success of the downstream training. Our approach takes a two-fold strategy of adjusting generation parameters to improve quality and filtering out low quality samples.

**Temperature**: Intuitively, adjusting the temperature is a common strategy to balance diversity and quality. Our framework also leverages this approach, using a relatively higher temperature during input generation to encourage diversity, and a lower temperature in other stages to promote quality. However, solely relying on temperature adjustment is insufficient to achieve the desired balance. SELF-GUIDE further employs two rounds of rule-based data filtering, one after the inputs are generated, and the other after the outputs are annotated. For clarity, we describe these two rounds together.

**Noise Filter**: We manually curate a list of noise terms, such as common salutations, greetings, and noise characters (e.g., $\backslash_-\backslash_-$ in the generated contents). If any noise term from this curated list appears in either the input or the output of a generated example, we discard the entire example. This filtering step ensures that the generated examples remain concise, focused, and free from irrelevant content or artifact patterns.

**Length Filter**: While the lengths of demonstrative examples may exhibit bias, we assume these examples already possess decent representativeness for a specific task in terms of length distribution. Based on this assumption, we further assume that the lengths of demonstrative examples follow a normal distribution for a specific task, and since most data points of a normal distribution fall within two standard deviations from the mean, we stipulate that the lengths of generated examples' inputs and outputs should also approximate a normal distribution with mean $\mu$ and variance $\sigma^2$ calculated from the demonstrative examples. Specifically, the lengths are required to be within the range $(\mu - 2\sigma, \mu + 2\sigma)$. Other attributes like semantic similarity of generated examples are computationally expensive and lack clear, intuitive definitions. Hence, we opt for the efficient and tractable length attribute under this normality assumption.

| Task ID | Task Description | Prompting *(Baseline)* | Few-Shot Finetuning | SELF-GUIDE *(Ours)* | Δ |
|---------|-----------------|------------------------|---------------------|---------------------|-----|
| | **Classification Tasks** | | | | |
| task1516 | NLI (IMPPRES) | 17.6 | 32.2 | 35.2 | 17.6 |
| task1529 | NLI (SciTail) | 8.5 | 48.9 | 54.5 | 46.0 |
| task1612 | Sentiment Class. (SICK) | 51.3 | 33.3 | 33.3 | -18.0 |
| task1615 | NLI (SICK) | 0.5 | 33.3 | 33.1 | 32.6 |
| task284 | Sentiment Class. (IMDB) | 90.0 | 71.9 | 82.2 | -7.8 |
| task329 | Coreferent Class. | 29.1 | 45.4 | 44.7 | 15.6 |
| task346 | Word POS Class. | 35.1 | 49.9 | 50.7 | 15.6 |
| **Avg** | **Metric: Exact Match** | **33.2** | **45.0** | **47.7** | **14.5** |
| | **Generation Tasks** | | | | |
| task1345 | Question Paraphrasing | 40.7 | 36.0 | 50.5 | 9.8 |
| task281 | Find Common Entity | 46.8 | 40.7 | 49.3 | 2.5 |
| task1562 | Question Paraphrasing | 29.5 | 48.6 | 59.3 | 29.8 |
| task1622 | Fluency Correction | 49.2 | 86.2 | 78.5 | 29.3 |
| **Avg** | **Metric: ROUGE-L** | **41.6** | **52.9** | **59.4** | **17.9** |

Table 1: For each task category (classification and generation), we randomly split the tasks into two halves, one half to tune the parameters for the "One Parameter Fits All" strategy, and the other half to test SELF-GUIDE's performance using this tuned set of parameters. We use the same decoding parameters and prompt template to evaluate the performance of the base model and the Self-Guided expert model on the held-out tasks. Δ is the performance difference between SELF-GUIDE and the Prompting baseline.

## 2.3 Quality Optimization: Parameter Tuning

We want SELF-GUIDE to generate training data matching to the desired distribution specified by an instruction and examples. This requires optimizing various hyper-parameters on labeled data points, including the initial number of generated inputs, the temperature for input generation, the temperature for output generation, finetuning parameters like training epochs, and so on. To achieve this, we tune the parameters on a set of *existing* tasks and their corresponding instructions, and then evaluate the model on a held-out dataset. We do so by searching parameters that maximize *worst task performance*, in order to identify parameters that are likely to be "good enough" for a broad set of tasks (Michel et al., 2021):

$$\max_{\theta} \min_{t \in \text{tasks}} \left( \text{performance}(\theta, t) - \text{ICL}(t) \right) \tag{1}$$

In Section 3 below, we demonstrate that our default set of parameters generally performs well with an absolute improvement of approximately 15% for classification tasks and 18% for generation tasks in the benchmark's metrics.

## 3 Experimental Setup

### 3.1 Datasets

To evaluate the effectiveness of SELF-GUIDE, we selected 14 classification tasks and 8 generation tasks from the Super-NaturalInstructions V2 benchmark. We randomly chose half of these tasks (7 classification and 4 generation) for hyper-parameter search, and the remaining half for evaluation (referred to as held-out tasks). Table 1 lists the held-out tasks. To find the optimal set of parameters as described in Section 2.3, we performed a random search over the temperature for input generation and output generation, as well as the number of generated inputs before filtering.

### 3.2 Base Model

We selected Vicuna-7b-1.5 (Zheng et al., 2023; Wan et al., 2024; Fan & Tao, 2024) as the foundational model for input generation, output generation, and fine-tuning for two primary

reasons: Firstly, having undergone fine-tuning on an extensive and diverse corpus of user conversations from ShareGPT, this model demonstrates robust cross-task instruction-following capabilities. However, despite its extensive training, its performance is not deemed satisfactory, as evidenced by its relatively low ranking of 73rd on the LMSYS Chatbot Arena Leaderboard (Chiang et al., 2024), prompting us to investigate opportunities for improvement.

We employ the same evaluation metrics as in the Super-NaturalInstructions V2 benchmark, namely Exact Match for classification tasks and ROUGE-L for generation tasks. Exact Match measures the exact string match between the predicted and ground truth labels, while ROUGE-L computes the longest common subsequence between the generated and reference texts. For a fair comparison, we use identical settings for our base model and Self-Guided model, including model size, batch size, decoding strategy (greedy search), and prompt template in Appendix A.1.2. During inference, we leverage the efficient VLLM framework proposed by Kwon et al. (2023). For supervised finetuning with teacher forcing, we employ the Hugging Face TRL library (von Werra et al., 2020) due to its flexibility and ease of use for transformer-based language models.

## 3.3 Baselines

SELF-GUIDE provides a method for executing tasks specified by prompts (which contain instructions and a small number of demonstration examples). We compare SELF-GUIDE against other methods for instruction following and in-context learning:

1. *Few-Shot ICL (Prompting)*: As our primary baseline, we compare against directly prompting the LM. This directly relies on the inherent instruction-following abilities of the model.
2. *Self-ICL*: We build on Self-ICL (Chen et al., 2023), which uses self-generated examples to improve zero-shot instruction following. We adopt Self-ICL to our few-shot setting by self-generating as many examples as can fit in the context window. We list this number of examples as *k* in Table 2.
3. *Few-Shot Finetuning*: We consider directly finetuning on the few demonstrations in each prompt. Prior work (Liu et al., 2022) shows this can be very effective.

We use the same base model (Vicuna-7b-1.5) as described in 3.2 for every baseline mentioned above.

## 4 Results and Analysis

Our main experiment results are shown in Table 1. Specifically, we report an absolute improvement of 14.5% for classification tasks and 17.9% for generation tasks in the benchmark's metrics. SELF-GUIDE demonstrates its effectiveness in guiding LLMs toward task-specific expertise, even in extremely data-limited situations. This highlights the potential of self-synthesized data in adapting LLMs for specialized tasks from a more scalable perspective.

To further analyze the reasons behind the improvement brought about by SELF-GUIDE, we conducted several analysis experiments to examine certain properties of SELF-GUIDE.

### 4.1 Comparing self-synthesized examples with gold few-shot examples

In Table 1, we see that SELF-GUIDE improves Vicuna-7b-1.5's ability to complete most tasks. Given that the SELF-GUIDE algorithm, in essence, involves finetuning on self-synthesized data, how much of this performance boost can be attributed to the synthetic data or to the learning algorithm?

To test this, we compare two ways of incorporating synthetic data into an LM: finetuning and in-context learning. For finetuning, we compare SELF-GUIDE with few-shot finetuning on three gold demonstrations. For in-context learning, we compare adding self-synthesized data into demonstration examples, i.e. Self-ICL, with in-context learning using only the three gold demonstrations.

| Task ID | 3-Shot Finetuning | SELF-GUIDE | 3-Shot ICL | Self-ICL (with $k$ syn. ex.) | Finetuning (on $k$ syn. ex.) | $k$ |
|---|---|---|---|---|---|---|
| **Classification Tasks** | | | | | | |
| task1516 | 32.2 | 35.2 | 17.6 | 4.5 | 33.2 | 60 |
| task1529 | 48.9 | 54.5 | 8.5 | 0.2 | 50.0 | 52 |
| task1612 | 33.3 | 33.3 | 51.3 | 34.1 | 33.3 | 39 |
| task1615 | 33.3 | 33.1 | 0.5 | 0.0 | 33.3 | 3 |
| task284 | 71.9 | 82.2 | 90.0 | 10.0 | 50.0 | 22 |
| task329 | 45.4 | 44.7 | 29.1 | 35.2 | 44.4 | 3 |
| task346 | 49.9 | 50.7 | 35.1 | 22.4 | 49.9 | 30 |
| **Avg** | **45.0** | **47.7** | **33.2** | **15.2** | **42.0** | **30** |
| **Generation Tasks** | | | | | | |
| task1345 | 36.0 | 50.5 | 40.7 | 48.8 | 50.4 | 7 |
| task281 | 40.7 | 49.3 | 46.8 | 36.8 | 51.5 | 28 |
| task1562 | 48.6 | 59.3 | 29.5 | 44.1 | 57.3 | 30 |
| task1622 | 86.2 | 78.5 | 49.2 | 50.9 | 78.4 | 42 |
| **Avg** | **52.9** | **59.4** | **41.6** | **45.2** | **59.4** | **27** |

Table 2: We compare in-context learning and finetuning on their ability to learn either from 3 manually written demonstrations or from a varying number of synthetically generated demonstrations. We find finetuning is better at leveraging synthetic data than in-context learning. When comparing Self-Guide with 3-Shot Finetuning, finetuning on self-synthesized data yields better performance than finetuning on gold-standard few-shot examples.

In Table 2, we see that using generated training data with finetuning improves over 3-shot finetuning on average for both classification and generation tasks (comparing the "SELF-GUIDE" and "3-Shot Finetuning" columns). On the other hand, comparing the "3-Shot ICL" and "Self-ICL" columns, we see that using generated training data with in-context learning helps only for generation tasks; for classification tasks, this strategy often leads to deteriorated performance. In Self-ICL for classification tasks, the prompted model tends to generate new input-output demonstrations rather than respond to the final input to be completed, leading to a significant performance decrease. We hypothesize that the model fails to segment the instruction from the demonstration examples, thereby failing to systematically classify the provided inputs according to the task definition.

## 4.2 Finetuning outperforms in-context learning on synthetic data

How much is the learning algorithm — finetuning versus in-context learning — responsible for the effectiveness of SELF-GUIDE? To test this, we compare in-context learning on as many self-synthesized examples as can fit in the context window of an LM ("Self-ICL") with finetuning on the same set of self-synthesized examples. We find in Table 2, compared to in-context learning, finetuning on the same generated examples achieves substantially better performance, with an average improvement of around 20 absolute percentage points across almost all tasks. Interestingly, finetuning on just a few self-synthesized examples (e.g. 7) can yield considerable performance gains, which highlights the data efficiency of SELF-GUIDE.

## 4.3 SELF-GUIDE aligns LMs with the correct label distribution in many cases

To demonstrate the high quality of data generated by SELF-GUIDE from a quantitative perspective, we analyze the distance between the output distributions of the models before and after SELF-GUIDE and the ground truth distribution for classification tasks.

Specifically, we compute the output distributions of the base model and the Self-Guided expert model on the same task. Considering that the outputs may contain irrelevant greetings or cases where the model directly refuses to answer, we collectively treat outputs without labels as irrelevant content and calculate the distribution across all labels and

irrelevant content. After obtaining the output distribution, we calculate the L1 distance between this distribution and the ground truth distribution.

| Task ID | Accuracy | | L1 distance | | Irrelevant Ratio | |
|---------|----------|------------|-------------|------------|------------------|------------|
| | Baseline | SELF-GUIDE | Baseline | SELF-GUIDE | Baseline | SELF-GUIDE |
| task1516 | 0.18 | 0.35 | 1.31 | 0.81 | 0.65 | 0.00 |
| task1529 | 0.09 | 0.55 | 1.79 | 0.23 | 0.90 | 0.00 |
| task1612 | 0.51 | 0.33 | 0.67 | 1.33 | 0.00 | 0.00 |
| task1615 | 0.01 | 0.33 | 1.97 | 1.33 | 0.98 | 0.00 |
| task284 | 0.90 | 0.82 | 0.14 | 0.33 | 0.07 | 0.00 |
| task329 | 0.29 | 0.45 | 0.85 | 1.09 | 0.25 | 0.00 |
| task346 | 0.35 | 0.51 | 0.97 | 0.16 | 0.28 | 0.00 |
| Avg | 0.33 | 0.48 | 1.10 | 0.75 | 0.45 | 0.00 |

Table 3: The L1 distance from the ground truth distribution, with lower values indicating better alignment. Notably, the Irrelevant Ratio column indicates the proportion of outputs deemed irrelevant, with the SELF-GUIDE model effectively reducing this ratio to 0 across all tasks.

From the results in Table 3, on average, the SELF-GUIDE model demonstrates a significantly lower average L1 distance compared to the baseline model while reducing the proportion of irrelevant content to 0. This indicates that the SELF-GUIDE model not only enhances the consistency of outputs with the true distribution but also effectively reduces irrelevant information in generated content, thereby enhancing the overall performance of the model.

## 4.4 SELF-GUIDE learns a non-trivial input-output mapping via self-synthesis

In this experiment, we investigate whether the improved performance of the LM under the SELF-GUIDE framework stems from merely learning trivial patterns like output formatting and label structuring, or also from gaining a better understanding of the task. We consider two scenarios: Rand-Baseline, where we randomize the labels in the demonstrative examples following Min et al. (2022b), and Rand-SELF-GUIDE, where we randomize the labels of the self-generated examples in the SELF-GUIDE approach. Specifically, for the Rand-Baseline, each original label in the demonstrative examples is replaced with a random label. For Rand-SELF-GUIDE, we randomize all the labels of the generated examples in SELF-GUIDE, finetune the base model on this randomized data, and evaluate it using the same prompt as in our main experiments.

From our results in Table 4, Rand-SELF-GUIDE outperforms the baseline suggesting that even with randomized labels, the self-generated examples provide valuable signals to the model. Additionally, the high ratio of irrelevant content produced by the baseline model, as detailed in Table 3, contrasts sharply with the SELF-GUIDE approach, which eliminates irrelevant outputs entirely, ensuring all outputs align with the expected label space. This demonstrates that self-generated examples can introduce basic patterns like output formatting and label structuring to language models. However, SELF-GUIDE's improvement gain over Rand-SELF-GUIDE further demonstrates that beyond learning superficial patterns, better supervisory signals during the SELF-GUIDE process enable the model to develop a deeper comprehension of the task itself. Finally, Rand-Baseline's worse performance compared to the baseline confirms that merely observing well-formatted outputs is insufficient - the model crucially requires proper supervision from high-quality demonstrations to truly grasp the task essence. Collectively, these contrasts reveal that the SELF-GUIDE method allows the model to simultaneously acquire shallow output patterns while leveraging the self-generated examples to model the task distribution, leading to a comprehensive boost in its capabilities on specific tasks.

| Task ID | Rand-Baseline | Baseline | Rand-SELF-GUIDE | SELF-GUIDE |
|---------|---------------|----------|-----------------|------------|
| task1516 | 18.7 | 17.6 | 33.2 | **35.2** |
| task1529 | 9.5 | 8.5 | 49.9 | **54.5** |
| task1612 | 40.6 | **51.3** | 33.3 | 33.3 |
| task1615 | 6.5 | 0.5 | **33.3** | 33.1 |
| task284 | 69.9 | **90.0** | 50.0 | 82.2 |
| task329 | 30.4 | 29.1 | 44.5 | **44.7** |
| task346 | 36.4 | 35.1 | 50.0 | **50.7** |
| Avg | 30.3 | 33.2 | 42.0 | **47.7** |

Table 4: Comparison of task performance on classification tasks between the SELF-GUIDE, the baseline model, Rand-SELF-GUIDE (where labels of self-generated examples are randomized in SELF-GUIDE), and Rand-Baseline (where labels of demonstrative examples are randomized). The best performance on each task is highlighted in bold.

## 4.5 Noise filter is crucial for classification tasks, while length filter is crucial for generation task

Ablation studies results below in Table 5 found that removing the ablation filter decreases classification accuracy by 4.1% while removing the length filter decreases generation accuracy by 3.7%. We think that the ablation filter is crucial for classification tasks as it removes outputs with superfluous content, retaining only the labels. For generation tasks, the length filter enhances performance by excluding lengthy or too short responses, improving the ROUGE-L score. These studies demonstrate the effectiveness of our rule-based filters.

| Task | w/o both | w/o ablation | w/o length | w/ both |
|------|----------|--------------|------------|---------|
| Generation | 55.3 | **59.6** | 55.7 | 59.4 |
| Classification | 44.4 | 43.6 | **47.7** | **47.7** |

Table 5: Comparison of task performance using different filters. The best performance on each kind of task is highlighted in bold.

In conclusion, we identify several key factors contributing to the efficacy of SELF-GUIDE. Initially, expanding and augmenting the training dataset with self-generated synthetic data derived from few-shot examples proves to be highly effective compared with manually collecting data. Second, SELF-GUIDE is aligned with the findings of prior research (Liu et al., 2022), which posits that finetuning surpasses in-context learning in terms of effectiveness. Third, our analysis reveals that finetuning synthesized data enhances the consistency of outputs with the true distribution and effectively reduces irrelevant information in generated content. Fourth, SELF-GUIDE enables LLM to develop a deeper comprehension of the task itself while learning superficial patterns. Last, filters are recommended both for the raw model and SELF-GUIDE, and the ablation filter is crucial for classification tasks, while length filter is crucial for generation tasks.

## 5 Related Work

Recent studies have demonstrated the efficient execution of language-based instructions by LLMs through fine-tuning instruction-based datasets. These datasets comprise pairs of language instruction commands and expected outcomes annotated by humans (Weller et al., 2020; Mishra et al., 2022). (Honovich et al., 2022) indicate that despite the noise present in LLM-generated datasets, they can still serve as effective training resources for instruction fine-tuning, implying that the parametric knowledge in pretrained LLMs contains, with potentially some transformation necessary, an inherent understanding of instructions. We extend this hypothesis, proposing that LLMs inherently possess the ability to understand arbitrary instructions and that this ability can be exploited to self-generate training data.

Manually crafted datasets have played a pivotal role in supervising and augmenting various NLP task systems (Rozière et al., 2023; Yuan et al., 2023). However, traditional manual annotation processes are often time-consuming, labor-intensive, costly, and non-scalable. Additionally, the possibility of human errors and the lack of domain expertise among annotators in complex tasks cannot be overlooked (Braylan & Lease, 2021; Kang et al., 2023). Recent works have tried to alleviate the human labor associated with training data collection by utilizing the power of stronger models as data generators (Patel et al., 2024; Wang et al., 2024; Song et al., 2024). These methods can be broadly categorized based on their reliance on seed data, the need for human curation, and stronger teacher LLMs. Viswanathan et al. (2023); Ge et al. (2024) combine retrieval methods and distillation to directly gather training data from external sources without seed data. Zhou et al. (2023) introduced LIMA, which generates possible responses using LLMs given existing instructions, and then curates them with human annotators. Li et al. (2023) automated the generation of instructions through back-translation on large external corpora, eliminating the need for seed data (Self-Alignment). However, these methods still require more powerful models or seed datasets to cultivate high-quality examples. Our work investigates the *self-generation* abilities of LLMs as a means of reducing the need for external resources compared to previous efforts.

Self-Instruct (Wang et al., 2023), which was mentioned in section 1, is the most relevant work to ours. They also use synthetic data self-generation; in their case, they focus on enabling data-efficient general-purpose instruction finetuning. Our proposed method explores the complementary question of how to use self-generation to make an instruction-finetuned model *even better* at executing the instructions specified for a given task.

## 6 Conclusion

In this paper, we propose SELF-GUIDE, an algorithm for large language models to follow task-specific instructions by internally producing synthetic training data and finetuning on this data. The improvement underscores the potential of our approach in enhancing LLMs' task-specific expertise, particularly in data-limited scenarios, and open avenues for exploring advanced techniques in autonomous model adaptation and continuous learning. We hope that our work can chart a path for future research in autonomous self-alignment and improvement mechanisms in AI systems, aligning them more closely with human intentions.

## 7 Limitations and Ethical Considerations

The primary limitation of our work is that we focus all of our experiments and models on English NLP tasks. Language technologies already promote systematic inequalities between languages and communities (Blasi et al., 2021; Zhao & Zhang, 2024; Zhao et al., 2024); SELF-GUIDE brings the possibility of exacerbating these inequalities. On the other hand, SELF-GUIDE also offers the tempting potential of improving LMs' ability to execute instructions specified in non-English languages to empower trustworthy non-English LLM-based Agents (Chen et al., 2024; Sun et al., 2024; Huang et al., 2023). We consider this to be an essential direction for future work, and we are actively pursuing this.

Another limitation of our work is that we've only shown that SELF-GUIDE works at improving a 7B-parameter model (Vicuna-7b-1.5). We believe that our approach could be used to improve the ability of arbitrarily large LMs (He et al., 2023; Hu et al., 2024) to follow task-specific instructions. However, due to budgetary restrictions, we were unable to experiment with larger models; we leave this as an important avenue for future work.

In terms of ethical considerations, SELF-GUIDE' potential to improve instruction following on specific tasks raises the risk of making it easier for people to specialize LLMs for malicious purposes. The open-sourcing of our code could amplify this risk. As Widder et al. (2022) points out (with the case study of deepfakes), open-source software can simultaneously increase the prevalence of harms while providing the community with the understanding and experience to manage these harms. Given SELF-GUIDE' potential for positive use, we believe that this algorithm and code are still worthy of releasing to the public.

## 8 Acknowledgments

This work was supported by the computing resources provided by the Microsoft Accelerate Foundation Models Research Program and the Tencent AI Lab Rhino-Bird Gift Fund. The authors would like to thank the reviewers for their valuable feedback. Additionally, we extend our gratitude to Professor Pengfei Liu at Shanghai Jiaotong University for his assistance with this work.

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

# A Appendix

## A.1 Prompt Template

### A.1.1 Input Generation Prompt Template

```
INPUT_GENERATOR_PROMPT_FOR_GENERATION = """
As an InputGenerator, your task is to generate
a new [input] based on the [instruction] and
some example [input].

Try your best to ensure that the new [input]
you generate is distinct from the provided
[input] while maintaining a diverse, detailed,
precise, comprehensive, and high-quality response.

Avoid generating a new [input] that is the
same as the provided [input].

[instruction]

{instruction}

Here are some high-quality [input] for the
[instruction]. These [input] can provide
you with very strict format requirements.
You should pay extreme attention to them!!!

Some high-quality [input]:

{high_quality_input_string}

These are some additional [input]. Their
formats and contents may not be accurate.
However, you may also refer to the content
of them.

Some low-quality [input]:

{low_quality_input_string}

After seeing example inputs, generate a new
[input]. Before generating the new [input],
ensure that you strictly adhere to the rules
of the new [instruction] and follow the
format of high-quality [input].

Prioritize the new [instruction] guidelines
to maintain consistency and quality.

Think twice before generating a new [input].
Only response the new [input] without any
other information.

[input]=
"""

INPUT_GENERATOR_PROMPT_FOR_CLASSIFICATION = """
As an InputGenerator, your task is to generate
a new [input] based on the [instruction] and
some example [input].

Try your best to ensure that the new [input]
you generate is distinct from the provided
```

```
[input] while maintaining a diverse, detailed,
precise, comprehensive, and high-quality response.

Avoid generating a new [input] that is the
same as the provided [input].

[instruction]

{instruction}

Here are some high-quality [input] for the
[instruction]. These [input] can provide
you with very strict format requirements.
You should pay extreme attention to them!!!

Some high-quality [input]:

{high_quality_input_string}

These are some additional [input]. Their
formats and contents may not be accurate.
However, you may also refer to the content of them.

Some low-quality [input]:

{low_quality_input_string}

After seeing example inputs, generate
a new [input] for which the expected
[output] is {conditional_label}. Before
generating the new [input], ensure that
you strictly adhere to the rules of the
new [instruction] and follow the format
of high-quality [input].

Prioritize the new [instruction] guidelines
to maintain consistency and quality.

Think twice before generating a new [input].
Only response the new [input] without any
other information. Note that the expected
[output] for the new [input] should be
{conditional_label}.

[input]=
"""
```

### A.1.2    Output Generation Prompt Template

```
OUTPUT_ANNOTATION_PROMPT_TEMPLATE = """
A chat between a curious user and an
artificial intelligence assistant.
The assistant gives concise answers
to the user's questions.
USER: The artificial intelligence
assistant only needs to
help annotate label.
The task is: {instruction}
ASSISTANT: Okay.
USER : [input] =
{input_1}
ASSISTANT : {output_1}
USER : [input] =
{input_2}
```

```
ASSISTANT : {output_2}
USER : [input] =
{input_3}
ASSISTANT : {output_3}
USER: [input] =
{new_input}
ASSISTANT:
"""
```

## A.2  Prompt Sensitivity Experiment

LLMs have been shown to be highly sensitive to prompt formats and minimal changes such as punctuation (Sclar et al., 2023b). This sensitivity can lead to drastic fluctuations in model performance, affecting its stability and reliability in real-world applications. Here, we introduce minimal prompt modifications. Although these modifications seem trivial from a human perspective, we will show that they greatly impact the base model's performance. However, Self-Guided final models prove robust, consistently outperforming the base model across various conditions. Specifically, we use the output generation prompt template (see Appendix A.1.2) to finetune the base model but make minimal disturbances to the few-shot examples of the prompt template when evaluating. Note that in the original format, there is an "=\n" following "USER : [input]". We change this conjunction between "USER : [input]" and the example input to ":", "\n\n", and evaluate the performance of the base model and the SELF-GUIDE expert model under this condition. Detailed results are shown in Table 6. We find that such simple changes, which may be overlooked by humans, can lead to catastrophic performance decreases in the raw model. On the other hand, SELF-GUIDE's performance exhibits remarkable stability and excellence.

| Category | Task ID | Raw Model | | | | SELF-GUIDE Expert Model | | | |
|---|---|---|---|---|---|---|---|---|---|
| | | =\n | : | \n\n | diff | =\n | : | \n\n | diff |
| **Classif.** | task190 | 27.0 | 21.1 | 19.8 | **7.2** | 66.7 | 66.7 | 66.7 | **0.0** |
| | task1529 | 8.5 | 11.0 | 29.5 | **21.0** | 61.6 | 54.9 | 53.1 | **8.5** |
| | task1612 | 51.3 | 50.9 | 48.8 | **2.5** | 42.6 | 42.4 | 42.8 | **0.4** |
| | task329 | 29.1 | 28.7 | 27.6 | **1.5** | 46.3 | 45.6 | 45.9 | **0.7** |
| | **Avg** | **29.0** | **27.9** | **31.4** | **8.1** | **54.3** | **52.4** | **52.1** | **2.4** |
| **Generat.** | task281 | 46.8 | 43.4 | 32.4 | **14.4** | 52.3 | 46.9 | 47.3 | **5.4** |
| | task1195 | 49.9 | 44.1 | 57.3 | **13.2** | 83.7 | 85.4 | 85.5 | **1.8** |
| | task1345 | 40.7 | 36.0 | 44.0 | **8.0** | 53.0 | 53.1 | 52.8 | **0.3** |
| | task1562 | 29.5 | 29.2 | 33.0 | **3.8** | 62.6 | 62.7 | 62.9 | **0.3** |
| | **Avg** | **41.7** | **38.2** | **41.7** | **9.9** | **62.9** | **62.0** | **62.1** | **2.0** |

Table 6: Sensitivity of models to prompt conjunctions on classification and generation tasks. We made minimal disturbances to the few-shot example section of the original prompt template, changing the conjunction between "USER: [input]" and the example input from "=\n" to ":" and "\n\n", and evaluated the performance of the raw language model (Raw Model) and the expert model generated by SELF-GUIDE (Expert Model) under these conditions. The table presents the average scores and differences for the two models across different task categories.

