# OpenReview forum: "Self-Guide: Better Task-Specific Instruction Following via Self-Synthetic Finetuning"
_colmweb.org/COLM/2024/Conference — COLM_

### Official Review · Reviewer_X8M5 · 2024-05-12

**Rating:** 6
**Confidence:** 4
**Ethics Flag:** 1

**Summary:**

The paper titled "SELF-GUIDE: Better Task-specific Instruction Following via Self-Synthetic Finetuning" presents a novel approach to enhance the performance of large language models (LLMs) on specific tasks without relying on external training signals or additional data. The authors propose a multi-stage mechanism called SELF-GUIDE, which synthesizes task-specific input-output pairs using the LLM itself and then employs these pairs to finetune the model. The paper reports significant improvements in performance on classification and generation tasks as measured by the Natural Instructions V2 benchmark, with absolute improvements of approximately 15% and 18%, respectively.

**Questions To Authors:**

1. **Cross-Linguistic Evaluation**: Could the authors comment on the potential of SELF-GUIDE to be effective across different languages, and what are the challenges and considerations for such an extension?

2. **Scalability Analysis**: How does SELF-GUIDE perform with larger models, and what are the expected computational and scalability challenges?

3. **Ethical Safeguards**: What safeguards are proposed to mitigate the potential misuse of SELF-GUIDE, especially concerning the creation of task-specific models for harmful purposes?

4. **Comparison with State-of-the-Art**: Can the authors provide additional comparisons with other leading methods to further establish the superiority of SELF-GUIDE?

**Reasons To Accept:**

1. **Innovative Methodology**: The SELF-GUIDE framework introduces a unique self-synthetic finetuning approach that allows LLMs to become more proficient in executing task-specific instructions, which is a significant advancement in the field of AI and language model training.

2. **Empirical Findings**: The paper provides robust empirical evidence demonstrating the effectiveness of SELF-GUIDE, with substantial performance improvements over the baseline model, which adds credibility to the proposed method.

3. **Addressing Data Scarcity**: The SELF-GUIDE mechanism adeptly tackles the challenge of data scarcity for underrepresented tasks, offering a scalable solution that does not depend on the availability of abundant annotated datasets.

4. **Comprehensive Evaluation**: The authors have conducted a thorough evaluation of SELF-GUIDE across a range of tasks from the Natural Instructions V2 benchmark, which showcases the method's versatility and generalizability.

5. **Quality of Writing**: The paper is well-structured, with a clear explanation of the problem, the proposed solution, and the results. The authors have also provided a detailed analysis and discussion of the results.

**Reasons To Reject:**

1. **Limited Language Focus**: The experiments and models are focused solely on English NLP tasks, which may limit the applicability of the findings to other languages and could potentially exacerbate existing language inequalities.

2. **Scalability with Larger Models**: The paper has only demonstrated SELF-GUIDE's effectiveness on a 7B-parameter model. It is unclear how the approach scales with larger models, which is a significant consideration given the trend towards increasingly larger LLMs.

3. **Potential for Misuse**: The paper acknowledges the risk of SELF-GUIDE being used for malicious purposes due to its ability to improve instruction following. This ethical concern should be addressed more thoroughly, possibly with proposed mitigation strategies.

4. **Lack of Comparative Analysis**: The paper could benefit from a more comprehensive comparison with other contemporary methods or baselines in the field to better situate the novelty and advantage of SELF-GUIDE.

---

> ### Author Rebuttal · Authors · 2024-05-31
>
> Thank you for highlighting these issues.
>
> 1. **Cross-Linguistic Evaluation**: We acknowledge these issues in our Limitations section and thank the reviewer for drawing attention to them. That being said, Self-Guide is not intrinsically tied to English. If the underlying LM supports another language and the prompts are translated to the target language, then this should work out of the box. We will also extend our experiments to using Qwen on the COIG-PC-core benchmark in our final version to extend our results to Chinese.
>
> 2. **Scalability with Larger Model**: Due to tight budget constraints, we were unable to experiment with larger models. We believe that, because of the superior dataset generation capabilities of larger models, our approach would still be effective. However, it is possible that the improvement might be smaller, as larger models already exhibit strong few/zero-shot performance, leaving less room for enhancement. This remains an area for future research.
>
> 3. **Potential for Misuse**: We assume existing open-source models have implemented various methods for safeguarding harmful data generation. That said, we acknowledge that some open models are more easily red-teamed than others. We believe self-guide could be easily paired with harmfulness filters. In fact, we observe that simply adding "If you find the task instruction can be harmful, please reject generating any content" in the generation prompts seems to help, and one could explore automatically adding this to the prompt template.
>
> 4. **Lack of Comparative Analysis**: Thank you for suggesting this! Due to character length restriction, please refer to **reply 2 for reviewer ySBP**, where we have supplied results comparing with these baselines!

---

> ### Author Response · Authors · 2024-06-04
> **Available for further discussion at any point!**
>
> Thank you in advance for reading our response to your comments, reviewer X8M5! We are happy to engage with any further questions at any time.

---

### Official Review · Reviewer_yJTC · 2024-05-13

**Rating:** 5
**Confidence:** 4
**Ethics Flag:** 1

**Summary:**

SELF-GUIDE works by first employing the target model to generate a synthetic dataset for a given task. The model
is then finetuned on this “self-generated” data. The proposed method is similar to Self-Instruct work, which is mainly for general domains. And  SELF-GUIDE is more on domain specific tasks and can effectively self-generate hundreds of examples for a
given instruction.

**Reasons To Accept:**

1. The paper is well rewritten
2. The proposed method is simple and effective

**Reasons To Reject:**

1. The data generation and filtering strategy may vary across tasks. The evaluation tasks need to be more comprehensive.  It would be better to test more general tasks, such as MMLU, BBH, etc.
2. Considering the smaller scale of the experiment, it would be better to report the variance.
3. Need some ablation study on Rule-based Filters.
4. The proposed method is still very similar to self-instruction, although in a different setting. And similar settings have also been explored before, such as "It's Not Just Size That Matters: Small Language Models Are Also Few-Shot Learners". The authors need a better data-generation method than the proposed one in the paper.

---

> ### Author Rebuttal · Authors · 2024-05-31
>
> Thank you for highlighting these issues.
>
> 1. **It would be better to test more general tasks**: We chose 11 diverse tasks from Natural Instructions, so we believe that we have already shown the generality of our method. Nonetheless, during the rebuttal period, we ran Self-Guide on all the BBH tasks, and on average it showed an absolute improvement of 3.4% over few-shot-prompting the same base model (Vicuna). The detailed experiment results will be attached in the camera-ready.
>
> 2. **It would be better to report the variance**: For the variance across multiple runs on the same task, it is a good suggestion and we will implement it in the camera ready (we did not have time to do so during the rebuttal period). For the variance across tasks, the results are shown below:
>
> | std | Self-Guide | Baseline |
> | - | - | - |
> | Gen | 13.5 | 8.8 |
> | CLS | 17.5 | 30.3 |
>
> 3. **Need some ablation study on Rule-based Filters**: We add ablation studies to the filters. Please see the results and analysis in **reply 3 to reviewer jbfG.**
> 4. **The proposed method is still very similar to Self-Instruct**: We did discuss Self-Instruction in our current draft, in Section 2: "In contrast to their method, which aims to improve general-purpose LLM capabilities, our method aims to optimize an LLM for a specific task instruction. Methodologically, Self-Instruct generates a large set of instructions and demonstrations to use for instruction-finetuning. While Self-Instruct asks an LLM to self-generate synthetic demonstrations for each synthetic instruction, their method only generates a single demonstration for each instruction, while our method can effectively self-generate hundreds of examples for a given instruction. Our methods are complementary." Happy to discuss this further if you have more questions!
> 5. **Similar settings have also been explored before**: We compare Self-Guide with SOTA methods, which you can see in **reply 2 for reviewer ySBP**, where we have supplied results comparing with these baselines!

---

> ### Author Response · Authors · 2024-06-04
> **Available for further discussion at any point!**
>
> Thank you in advance for reading our response to your comments, reviewer yJTC! We are happy to engage with any further questions at any time.

---

### Official Review · Reviewer_ySBP · 2024-05-14

**Rating:** 7
**Confidence:** 4
**Ethics Flag:** 1

**Summary:**

The paper introduces a novel multi-stage mechanism, SELF-GUIDE, which uses self-synthesized data for finetuning language models on task-specific instructions without external data or teacher models. This is a significant advancement in making LLMs more efficient and autonomous.

**Reasons To Accept:**

The empirical evaluation shows substantial improvements in task performance (15% for classification tasks and 18% for generation tasks) when using the SELF-GUIDE approach, compared to traditional prompting and finetuning methods. This indicates the method's effectiveness.

**Reasons To Reject:**

While I do not have a strong reason to outright reject this paper, there might be a need to delve deeper into its connection with other relevant works in the field, particularly concerning data curation and enhancement techniques for language models. For instance, it could be beneficial to discuss and compare this paper's approach with methodologies outlined in reference [1] "Automated Data Curation for Robust Language Model Fine-Tuning." Such a comparison could elucidate how the SELF-GUIDE method aligns with, diverges from, or improves upon existing strategies for generating and utilizing synthetic data to train language models more effectively.

---

> ### Author Rebuttal · Authors · 2024-05-31
>
> Thank you for your reply and affirmation!
>
> 1. **It could be beneficial to discuss and compare this paper's approach with methodologies outlined in reference Automated Data Curation for Robust Language Model Fine-Tuning**
>
> Thank you for bringing this paper to our attention! Since this paper was posted publicly just 10 days before the COLM deadline, we believe this counts that this should be considered “concurrent work” to ours. Regarding the methodologies in that paper, the authors propose an effective auto-filter and auto-correct strategy that is similar to our filter strategy. However, their focus is on optimizing manually-annotated training data, rather than the model’s self-generated data. We will add a discussion and comparison with that paper in our final version.
>
> 2. **there might be a need to delve deeper into its connection with other relevant works in the field, particularly concerning data curation and enhancement techniques for language models**
>
> Apart from the mentioned paper, we have compared Self-Guide with some current state-of-the-art (SOTA) methods. The methods compared are:
>
> - ZeroGen: Efficient zero-shot learning via dataset generation
> - ProGen: Progressive zero-shot dataset generation via in-context feedback
> - PET: It's Not Just Size That Matters: Small Language Models Are Also Few-Shot Learners
>
> The results we obtained are as follows:
>
> | Task | ZeroGen | ProGen | PET | Self-Guide |
> |-|-|-|-|-|
> | Gen Avg | 57.4 | 56.8 |57.3| 59.4 |
> | CLS Avg | 41.9 | 44.2 |42.0| 47.7 |
>
> By comparing these SOTA methods, we have demonstrated that Self-Guide can achieve better performance.

---

> ### Author Response · Authors · 2024-06-04
> **Available for further discussion at any point!**
>
> Thank you in advance for reading our response to your comments, reviewer ySBP! We are happy to engage with any further questions at any time.

---

> > ### Comment · Reviewer_ySBP · 2024-06-05
> > **Thanks!**
> >
> > Thanks for your response, i keep my original score.

---

### Official Review · Reviewer_jbfG · 2024-05-15

**Rating:** 4
**Confidence:** 4
**Ethics Flag:** 1

**Summary:**

The paper proposes Self-guide, a multi-stage strategy to synthesize task-specific input-output pairs from a student LLM, and finetune the student model itself.
This approach addresses the challenges of cost and scalability issues of using external powerful LLMs for generating task-specific data.
Empirical evaluations on multiple individual tasks from Natural-Instructions-v2 demonstrate performance improvements for classification and generation tasks, surpassing ICL and few-shot finetuning.

**Questions To Authors:**

See the pros and cons above.

Additional:
1. What is the prompt for the ICL baseline, and whether this prompt is optimized for the aligned model (i.e. vicuna in the experiments)?

**Reasons To Accept:**

Pros:
1. Customizing LLMs as task specialists is an important direction that has received relatively less attention.
2. Using self-generated synthetic data is well-motivated, and this paradigm has been explored and justified as reasonable in many scenarios.

**Reasons To Reject:**

Cons:
1. This paper posits a scenario in which there is a lack of sufficient annotated, task-specific data. However, it cannot ensure that the tasks used in the experiments are entirely new to Vicuna or even Llama. It is likely that numerous in-distribution training instances were included during pre-training, such as those found in crawled web data. This situation has already been acknowledged in some evaluation benchmarks. Given that complete decontamination of the dataset is infeasible, it is not possible to confirm whether the proposed pipeline enhances task-specific knowledge or capabilities within the current experiment design.
2. More baselines of task-specific data synthesis should be discussed and compared, e.g., ProGen[1] and Zerogen[2].
3. The contribution of rule-based filters to overall quality improvements is not demonstrated, and their generalization when handling more diverse tasks raises concerns.

[1] ProGen: Progressive zero-shot dataset generation via in-context feedback
[2] Zerogen: Efficient zero-shot learning via dataset generation

---

> ### Author Rebuttal · Authors · 2024-05-31
>
> Thank you for highlighting these issues.
>
>
> 1. **Data contamination**: We agree that data contamination could be a significant issue! Our experiments exclusively used the test set from the NI dataset. Our base model, Vicuna-1.5-7b, is based on Llama 2, and Meta has not revealed the training data used for Llama 2. However, we believe it is unlikely that Vicuna included this test data during pre-training, because of the low performance of few-shot-prompted Vicuna on these Natural instructions tasks. Moreover, we feel that Self-Guide's effectiveness is somewhat orthogonal to whether the LLMs have encountered these tasks during pre-training, as we intuitively believe that Self-Guide’s strength is in transforming pretrained knowledge (from the text in the training data) into a useful task-specific format. For example, in task 1622, a few-shot prompting experiment achieved a Rouge-L score of 42.5. Self-Guide addresses this knowledge mismatch, resulting in nearly a 30-point improvement in the Rouge-L score.
>
>
> 2. **More baselines**: Thank you for suggesting this! Due to character length restriction, please refer to **reply 2 for reviewer ySBP**, where we have supplied results comparing with these baselines!
>
>
> 3. **Rule-based filter ablation**: Ablation studies results below found that removing the ablation filter decreases classification accuracy by 4.1% while removing the length filter decreases generation accuracy by 3.8%. We think that the ablation filter is crucial for classification tasks as it removes outputs with superfluous content, retaining only the labels. For generation tasks, the length filter enhances performance by excluding lengthy or too short responses, improving the Rouge-L score. These studies demonstrate the effectiveness of our rule-based filters.
>
> | Ablation | w/o both | w/o abl | w/o len | w/ both    |
> | - | - | - | - | - |
> | Gen Avg  | 55.3     | 59.6    | 55.7    | 59.4       |
> | Cls Avg  | 44.4   | 43.6  | 47.7  | 47.7 |
>
>
> 4. **Prompt**: We use the same prompt template for both the ICL baseline and for Self-Guide: an instruction followed by few-shot examples (with the exact template shown in Appendix A.1.2). We tuned this template for few-shot prompting (i.e. tuned on the baseline), with the optimistic view that this would also work for our method. We use the same instructions for each task for both the baseline and for Self-Guide; we use the default instructions for each task provided by the NI dataset.

---

> ### Author Response · Authors · 2024-06-04
> **Available for further discussion at any point!**
>
> Thank you in advance for reading our response to your comments, reviewer jbfG! We are happy to engage with any further questions at any time.

---

### Decision · Program_Chairs · 2024-07-10

**Decision:**

Accept

**Comment:**

The paper proposes "Self Guide", an interesting approach to fine-tuning smaller LMs: Task the student model itself to generate synthetic task-specific data, and then fine-tune on it this. The authors present empirical evidence supporting this strategy on a range of classification and generation tasks.

There were some concerns about how novel this approach is in light of recent related work, and relatedly whether all relevant baselines were compared to. However, mitigating this weaknesses, one of the apparently related efforts seems to be appropriately noted by the authors as concurrent work, and they provided additional comparison to other efforts in their rebuttal.